# Experience of Stress Assessed by Text Messages and Its Association with Objective Workload—A Longitudinal Study

**DOI:** 10.3390/ijerph17030680

**Published:** 2020-01-21

**Authors:** Bozana Arapovic-Johansson, Charlotte Wåhlin, Jan Hagberg, Lydia Kwak, Iben Axén, Christina Björklund, Irene Jensen

**Affiliations:** 1Unit of Intervention and Implementation Research for Worker Health, Institute of Environmental Medicine, Karolinska Institute, 171 77 Solna, Sweden; Charlotte.wahlin@regionostergotland.se (C.W.); jan.hagberg@ki.se (J.H.); lydia.kwak@ki.se (L.K.); iben.axen@ki.se (I.A.); christina.bjorklund@ki.se (C.B.); irene.jensen@ki.se (I.J.); 2Occupational and Environmental Medicine Center, and Department of Health, Medicine and Caring sciences, Unit of Clinical Medicine, Linköping University, 581 83 Linköping, Sweden

**Keywords:** work stress, objective data, workload, intra-individual variability, new technology

## Abstract

Exploring stress trajectories in detail and over a long time may give valuable information in terms of both understanding and practice. We followed a group of primary health care employees in a randomized controlled trial. The objective was to describe their experience of stress, explore the intra-individual variability and examine the association between the experience of stress and the objective workload. Weekly text messages with a single item stress question were distributed in two time series: 12 weeks at the beginning of the trial and 26 weeks after the 6-month follow up. Aggregated objective data about workload were collected from their administration office and related to stress levels. There was a seasonal variation, with higher stress during the fall than in spring and summer. The analysis comparing high and low stress subgroups showed that the stress trajectory of a high-stress subgroup was different from that of a low-stress subgroup. Individuals with high exhaustion scores had higher odds of belonging to a subgroup of individuals with high intra-individual variability in stress experience. The objective workload was measured in two ways and was strongly associated with the stress experience. We found that the lower the productivity, the higher the feeling of stress.

## 1. Introduction

Periods of prolonged stress can have serious consequences for an individual’s health [1] and can consequently lead to both suffering for the individual and high societal costs related to medical treatment, sick leave and productivity loss [2]. It is therefore important to develop feasible methods for the early identification of individuals and groups at risk of prolonged stress.

Typically, stress level assessment is carried out by questionnaires distributed at baseline and at follow up [3]. Questionnaires consisting of several items are preferable from a psychometric point of view, as factor analysis can be used to measure their internal structure [4]. However, they are not always suitable for frequent and prolonged monitoring and may sometimes be burdensome for participants. Collecting data by weekly messages provides more information about the trajectories of stress (given that the single items used are validated).

Even when prolonged, the stress experience probably fluctuates to some degree, not only over the day but also over days, weeks, months or years. For this reason, exploring stress trajectories in more detail and over a prolonged period may give a valuable information in terms of both understanding and practice. Knowing when to expect periods of high and low stress in a particular population enables resource planning and allocation.

New technology brings new possibilities. Mobile phone messaging is used for a range of purposes such as communicating results of medical investigations [5]. It can also be used to support health behaviors for preventive health care [6]. Moreover, weekly text messages (short message service, SMS) are a useful method for frequent monitoring [7]. Earlier research has shown that they have high response rates, are user-friendly [8] and inexpensive [9]. The method opens the door for gathering information from populations that may otherwise be difficult to reach, for example rural communities [10], low resource settings [11] but possibly also those with limited computer access or highly mobile individuals. Moreover, the method reduces data handling errors as the responses are registered in the data file directly [9].

In this study, we use weekly text messages to monitor the stress experience at work, and we relate this experience to objective organizational measures of workload. A recent systematic review of smartphone-based self-assessment of stress [12] reported that 35 studies had used smartphone-based assessment of stress, but only six of them measured self-assessed stress in the workplace. Of these six studies, four were conference papers and only two were published in a peer-reviewed journal [13,14]. These two published studies had small samples (30 and 35 participants respectively) and used a non-validated self-assessment question about stress. Furthermore, the aim of the first study [13] was to detect stress in an automatic manner, using smartphones’ built -in accelerometer and relating that data to the self-reported stress levels during working hours (three times a day during eight weeks). The second study [14] was also about using features from smartphones to asses stress experience. Smartphone sensors (recording physical activities, communication data, etc.) recorded data several times a day for four months and chest belt was used to measure the heart rate variability (HRV) during the night. The user was also asked to fill the questionnaire and answer the stress question before going to sleep. The aim was to “find appropriate feature set for smartphone and HRV in order to build logistic regression models for discriminating stress levels”. In conclusion, there is to our knowledge no previous study that has assessed perceived stress by weekly SMS using a validated question in a work context over such a long time-period as this study, nor related that weekly data to the objective organizational measures of workload.

In a study of individuals with back pain [8] researchers found that measuring a fluctuating construct on few occasions may identify a stability which does not actually exist or may give a distorted picture of the trajectory. The experience of stress is also fluctuating, as a stress response implies a comparison of present sensory information with stored brain information [15] and sensory information is constantly fluctuating. Therefore, frequent monitoring of stress levels may give a more detailed picture of both the individual and the group-level stress experience and its variations in time than less frequent surveys. Even individuals suffering of chronic stress have some variation in their daily/weekly experience of and response to episodic stress. The patterns of this variation may be important to understand and can possibly be captured by frequent text messaging.

The first aim of this study is to describe the trajectories of stress experience over time in a group of Swedish primary health care employees. The reports from the Swedish Work Environment Authority have shown that employees in the Swedish health care sector experience high levels of stress [16,17]. However, little is known about stress trajectories in this population. Increasing our understanding can enable employers to work proactively and in collaboration with employees to prevent recurrent periods of high stress and to plan for recovery after intensive work periods.

The second aim of this study was to examine the association between the stress experience and objective organizational measures of workload. High workload is one of the factors that may influence how we experience stress [18]. Therefore, measuring and acknowledging workload and allocating resources accordingly is one possible way of preventing stress. The most common way of measuring workload is by subjective measures. However, the use of subjective measures of workload raises the question of individual differences in perceptions of high or low workload, because positive and negative emotional states may influence people’s answers [19]. One way of avoiding this problem is to measure the objective workload and to relate it to employees’ self-reported stress [20]. To the best of our knowledge, little research has yet been undertaken into objective workload measures and work-related self-reported stress, and those findings which do exist are inconsistent. For example, the relationship between long working hours, health and well-being is ambiguous. The effects of long working hours depend on which aspects of physical or mental health are examined, as well as on cultural differences, occupation, working conditions, age, etc. [21]. Ganster et al. conclude that the relationship between working hours and psychological well-being is unclear. Relating the number of working hours to productivity data is one possible way to further develop objective workload measures because this gives an indication of the relative workload, i.e., adjusted to working hours.

Work demands (stressors) are often measured as self-reported perceived workload. The objective measures of workload, as used in this study, are not meant to be a better measure. Rather, they are seen as an important complementary measure to self-perceived workload. People differ in how much work they can manage, however an overemphasis on individuals’ “perceptions” may be both stigmatizing [22] and result in mostly individual interventions being recommended, i.e., resilience building, mindfulness, etc. Resilient individuals may cope better with stress and not end up being on sick leave, but that does not necessarily imply that the workload levels or the working environment is healthy. It may just mean that many individuals survive in a non-optimal environment. Using detailed longitudinal data to examine the group level associations between the objective workload and the experience of stress can remove the overemphasis on individuals and widen the discussion on stress management, in line with the research on organizational constraints and their importance for employee health and well-being [23]. Furthermore, different aspects of the work environment have an impact on employee wellbeing, but also on the quality of patient care. For example, a previous study has shown that high workload was associated with less individualized patient care [24]. In other words, less consideration was given to the patients’ needs and preferences due to high workload. Moreover, Aiken et al. [25] showed that hospitals with better care environments had lower risk of patient death and failure to rescue than hospitals with poor care environments. A literature review [26] showed that nurse staffing (most often measured by calculating nurse to patient ratio) was associated with better patient outcomes both in general care population and in intensive care units. More detailed inquiry into the association between objectively measured workload and employee stress and health should therefore be of interest for the hospital managements, primary health care managements and health care policy decision makers.

The third aim of this study was to examine the intra-individual variability of the weekly stress experience. Environmental factors such as workload influence stress levels, but proneness to stress and its consequences are also affected by individual risk factors such as genetic predisposition [27], past history with chronic or acute stress [28] and cognitive and behavioral patterns such as hardiness [29] and coping style [30]. In work-related stress research, the coping pattern called over-commitment has been shown to be an important risk factor for stress-related ill-health [31]. Measuring inter-subject differences is therefore important but not sufficient, because large intra-individual variability may bias the results of the inter-subject analysis, particularly if measured on one occasion only [32].

Where assessing the stress experience is concerned, the estimated average for a particular time period may not be the only important measure. The intra-individual variability in repeated measures during the time series can also provide important information about an individual, such as the rate at which individual stress experience (due to possible episodic stressors) changes. This can be relevant with regard to the discrimination of subgroups or for discovering important associations with other constructs. For example, a fast speed of change in the stress experience could be an indication of a reduced tolerance of further stress, which is one of the characteristics of exhaustion disorder [33]. In the present study, we employ explorative analyses to examine whether high variability in speed of change adds any information to the intra-individual standard deviation of a time series in this population. The idea is to explore whether a speed of change in the stress experience is different for people who score high on depression or exhaustion, or individuals with the work-related coping pattern of over-commitment, which is known to be a risk factor for exhaustion [31]. As a comparison, some studies [34] suggest that work-related stress and depressive symptoms can be linked to different methylation patterns of the brain-derived neurotrophic factor gene (BDNF). In other words, individuals with depression and exhaustion may differ in bio-physiological markers. Similarly, it is possible that they have different patterns in the speed of change of their self-reported stress experience. We are not implying that depression and exhaustion are “causing” these patterns, but simply examining if they are differentially associated with these patterns. The role of stress in the development of depression and exhaustion is well known, however, once depressed or exhausted individuals will continue to experience and react to episodic stress. The pattern of these reactions is interesting both theoretically and clinically, regardless whether depression and exhaustion are seen as separate constructs or if exhaustion (and burnout) is seen as just a subtype of depression, as suggested by some researchers [35]. For example, chronic depression and major depressive disorder do not have the same clinical picture, progress or prognosis and are not clinically treated in the same way even though they are both regarded as depressive disorders.

## 2. Materials and Methods

This longitudinal open cohort study is part of the Stress prevention at work (SPA) study, a two-armed randomized controlled trial (RCT) in a Swedish primary health care setting. Details of the RCT are reported in a previous paper [36]. Data in the SPA-study were collected over 12 months using questionnaires (baseline, 6- and 12-month follow-up), SMS, focus groups, individual interviews and register data. SMS-data were collected during two time-series: SMS-series 1 (12 weeks) starting after baseline in October 2013, ending in December 2013 and SMS-series 2 (26 weeks) starting six months after the baseline (March 2014) and ending in September 2014, just before the 12 month follow up. This study uses SMS data, register data and data on depression, exhaustion and over-commitment from the baseline and 6 months questionnaires. For detailed information, see the flow chart (Figure 1). For distribution of text messages and collection of answers we used SMS-Track Questionnaire software system [37] which is a system developed for research purposes and previously tested in several research settings [8].

### 2.1. Participants

The sample in the current study consisted of primary health care employees who participated in the SMS part of the SPA project. The total sample in the study consisted of 118 employees at the baseline and 123 employees at the 6-month follow up, as this was an open cohort. For research aims I and II, the sample consisted of those employees who responded to the SMS at least 80% of the weeks, regardless of whether they completed the questionnaires (SMS-series 1: n = 90 (76%); SMS-series 2: n = 82 (67%)). For research aim III, the sample consisted of those employees who responded to the SMS at least 80% of the weeks and completed the questionnaires (SMS-series 1: n = 80 (68%); SMS-series 2: n = 80 (65%)).

### 2.2. Measures

Data were collected on stress, exhaustion, depression, over-commitment, quantitative workload and socioeconomic status.

#### 2.2.1. Stress

A single item stress question (SISQ) from QPSNordic34+ [38] was administered weekly by means of a text message system, SMS Track^®^ (SMS-Track ApS, Esbjerg, Denmark) [37]. We used the Swedish validated version of the SISQ: “Stress means a state in which a person feels tense, restless, nervous or anxious or is unable to sleep at night because his/her mind is troubled all the time. Do you feel this kind of stress these days?” [38,39]. The responses were recorded on a 5-point Likert scale from (1) “not at all” to (5) “very much”. Automatic text messages were sent every Friday and an automatic reminder was sent to non-responders on the following Sunday [40].

#### 2.2.2. Exhaustion

Exhaustion was measured by the validated Swedish version of the Oldenburg Burnout Inventory-OLBI [41,42]. It has eight questions, for example “During my work, I often feel emotionally drained” The answers are given on a 4-point Likert scale (ranging from 1 = strongly disagree, to 4 = strongly agree). Possible scores are between 8 and 32. The higher the score, the more exhaustion is indicated (0–17.59 non-exhaustion; 17.60–21.99 means mild exhaustion and 22–32 is indicative of severe exhaustion). Cronbach’s α for our sample was 84 at the baseline and.82 at 6 months measurement.

#### 2.2.3. Depression

Symptoms of depression were measured by the seven questions on depression from the Swedish validated version of the Hospital Anxiety and Depression Scale-HAD [43,44]. Example of questions: “I look forward with enjoyment to things”. Answers were given on a 4-point scale, from 0 (Not at all) to 3 (Nearly all the time). Cut off scores indicate 0–7.99 for non-depression, 8–11 indicating mild depression and 11.01–21 indicating severe depression. Cronbach’s α was 86 at the baseline and 85 at 6 months measurement.

#### 2.2.4. Over-Commitment

Over-commitment to work was measured with a subscale of the Effort Reward Imbalance questionnaire [45] consisting of six questions, for example “People close to me say I sacrifice too much for my job”. Answers were given on a 4-point scale (ranging from 1 = strongly disagree to 4 = strongly agree). In the most comparable studies with health care employees, the mean score is between 11.79 and 15.95 [45], and the higher the scores the higher the over-commitment. Cronbach’s α for our sample was 81 at the baseline and 86 at 6 months measurement.

#### 2.2.5. Objective Organizational Measures of Quantitative Workload

In the Swedish health care sector, productivity output is usually measured as number of visits, surgical procedures, patients, etc. [46]. In this study, we used the ratio between the time unit (working hours) and the productivity output as a measure of objective workload. This productivity ratio enables one to compare workload over time (potentially also between units), just as the ratio between resources and results gives a measure of efficiency and enables one to compare efficiency [46].

Data about objective workload (hours worked, number of tasks, number of patient visits, number of administrative tasks and number of phone calls answered) were collected every month from the central administration office of the county council. These data were collected from each health care unit separately. This means that every separate unit´s objective data was related to their employees’ stress experience and in that way the type of unit was controlled for. The data were used in the analysis both as raw data and computed into four quantitative workload ratios. These were the ratios between the monthly total number of working hours as a nominator and the following denominators: (a) number of tasks, (b) number of phone calls answered, (c) number of patient visits, and (d) number of administrative tasks. The “number of tasks” is the organization’s own measure of the total number of monthly tasks, consisting of all their tasks, i.e., not only the sum of phone calls, number of visits and number of administrative tasks.

Ethical approval: Stockholm regional Ethical Review Board has approved this project. 2012/2200–31/5.

### 2.3. Statistical Analysis

We are modeling change over time for the SMS 1 series and the SMS 2 series separately as there was a three-month gap between these series. Furthermore, due to the natural turnover of staff (parental leave, studies, etc.) there was a small variation in the number of employees participating in the two series.

#### 2.3.1. The Analysis Pertaining to the First Aim: To Examine and Describe the Experience of Stress Over Time in a Group of Swedish Primary Health Care Employees

The sample here consisted of all employees who responded to the SMS at least 80% of the weeks, regardless of whether they completed the questionnaires. The mean of the SISQ was computed for all the participants in SMS series 1 (12 weeks) and SMS series 2 (26 weeks). Based on the present sample data (frequency tables), the cut off for the dichotomization of high and low stress sub-groups was set at the upper quartile (SISQ mean ≥ 3.67 = high stress subgroup in SMS series 1 and SISQ mean ≥ 3.19 = high stress subgroup in SMS 2). The reason for this was that we were interested in the trajectories of the most stressed employees. Because we had a 1-5 scale, the cut off had to be higher than three. The quartile cut-off gave the acceptable number of individuals (25 of 90 employees in the SMS 1; 21 of 82 employees in SMS series 2). We then computed means and standard deviations for every week for subgroups with high and low stress for SMS series 1 and 2 separately. As there was a small variation in the number of employees participating in the two SMS series (due to the natural turnover described above) and a three months gap between the two series, we could not treat the two SMS series as one.

#### 2.3.2. The Analysis Pertaining to the Second aim: The Associations between the Experience of Stress and the Group-Level Objective Organizational Measures of Quantitative Workload

The objective workload data were obtained monthly per health care unit and aggregated for the total sample. It was not possible to access the objective data on a weekly level and therefore the monthly SISQ mean was calculated for every individual from their weekly stress ratings.

One way to examine the association between the stress experience and objective organizational measures of quantitative workload, would be to compare the aggregated monthly SISQ scores for the total sample with the monthly objective raw data for the total sample, using a measure of strength of association (Pearson correlation). However, ecological correlations, i.e., correlations of means at group level, can give inflated (too high) estimates of correlations [47]. Therefore, we used generalized estimating equations (GEE) to examine the association between the quantitative monthly workload at health care unit level and the monthly experience of stress at individual level. However, the results are to be interpreted at a group level even if GEE in the analysis uses the individual level data on stress experience.

We calculated the workload ratios described in the Measurement section and used GEE both with the raw workload data and with the described workload ratios. Separate analyses were carried out for the SMS 1 and 2 series. In GEE we chose the autoregressive correlation type for two reasons: (a) the autoregressive-1 (AR (1)) correlation type performed better than the unstructured correlation matrix because the Quasi likelihood under Independence Model Criterion (QIC) was lower and (b) given the temporal nature of the repeated measurements, it seemed reasonable to try the AR (1) structure.

#### 2.3.3. The Analysis Pertaining to the Third Aim: To Describe the Intra-Individual Variability (i.e., Patterns of Fluctuation) in the Experience of Stress

Intra-individual variability is usually examined by means of standard deviation of a time series. However, this neglects the rate at which individual experience of stress changes [48]. This means that two individuals with the same standard deviation may nevertheless differ in variability, as one can be more “reactive”, with a higher rate of change from week to week. We therefore analyzed intra-individual variability in two ways: by means of standard deviation of a time series and by analyzing the speed of change:(A)We are not aware of any previous studies that have identified cut off scores for high and low stress subgroups based on a time series. Therefore, we first computed means (M) and standard deviations (SD) of all the stress scores for all participants in the SMS series 1 and SMS series 2 separately. Based on the frequency tables of these means and standard deviations, we selected four stress sub-groups: LL (low M/low SD), LH (low M/high SD), HL (high M/low SD), and HH (high M/high SD). When we chose the upper quartile as cut off for dichotomization, only a few persons fell into the HH group. We therefore chose the upper tertile for dichotomization of high/low to obtain more individuals in the HH group. Table 1 displays the means and the standard deviations used in the formation of these sub-groups.(B)Secondly, the intra-individual variability was calculated by the ordering of observations. For some individuals, high and low scores can follow each other in quick succession. The speed of change can be described by calculating so-called first difference (called first derivative in [48]). In other words, some individuals do not change much between two occasions while others have rapid increases or decreases in scores [48]. The first difference is the value of the measurement at a time t minus the value of the measurement at the previous time. The four subgroups were formed as described under A), but with the cut off for the SD of the first difference = 1.115 for SMS series 1 and SD = 1.098 for the SMS series 2.

For both A and B, we compared subgroups with regard to background data (age, years working in the organization, overtime work, overall health) as well as the baseline levels for over-commitment, exhaustion and depression. As male study sample was too small, we could not examine the differences based on sex. For SMS series 2, the 6-month follow up data was used as the background data. Multinomial regression was used in the analysis of both (A) and (B), not to draw any conclusions about the cause and the effect, but simply to compare the formed subgroups regarding the background data and the levels of over-commitment, exhaustion and depression at the start of the each SMS series. All analyses were performed with IBM-SPSS Statistics v.25 (Karolinska Institutet, Stockholm, Sweden)).

## 3. Results

Table 2 displays the background data of the study population. The mean age was 46 years and 84% of the study population were women. Most of the employees had university or higher academic education (81% in SMS series 1 and 85% in SMS series 2).

### 3.1. The Experience of StressOver Time

Figure 2 is a visual representation of the SMS mean for the SMS series 1 and 2. The exact mean and standard deviation values are displayed in Appendix A (Table A1).

The results show a variation in overall stress levels, with higher stress during the fall than in spring and summer. The variation in SMS series 1 is also noteworthy because the stress curve declines in weeks 5–6 (fall holidays) and goes up in weeks 11 and 12 (i.e., before Christmas). The highest stress levels are shown for weeks 2 and 3 (mid-October), 7 (mid-November) and 12 (mid-December).

Variations were also observed during the summer. After week 19 (the end of July), the stress curve slowly climbs and by week 25 (the beginning of September 2014) stress is at the same level as it was at the beginning of October 2013.

It was also apparent that the stress curve of the high-stress subgroup has a different shape than those of the total group and the low-stress subgroup. While stress declines after week 7 in the low-stress subgroup in SMS 1 series (see Figure 2), it continues to rise in the high-stress subgroup. The high-stress subgroup also seems to have a more fluctuating pattern than those of the total group and the low-stress subgroup. However, it is important to notice that SMS 1 and SMS 2 series do not overlap, i.e., there is a three months gap between them.

### 3.2. The Experience of Stress and the Objective Measures of Quantitative Workload

The GEE analysis showed that, for SMS series 2, quantitative monthly workload at group level is associated with the monthly experience of stress on a group level (*p* < 0.001). No such association was observed for SMS series 1 (parameter estimates not displayed).

Table 3 displays parameter estimates for SMS series 2 for both the raw objective data and for the ratios. The stress increases with higher numbers of patient visits, administrative tasks, telephone calls answered and the total number of tasks, as well as the total number of hours worked.

The associations between the four ratios and the stress experience give somewhat different results. We found that the higher the four monthly ratios, i.e., the fewer the tasks accomplished at group level per hour worked, the higher the monthly stress. In other words, the lower the productivity, the higher the feeling of stress. For example, in this sample the minimum for the ratio hours worked/total number of tasks is 0.668 and the maximum is 0.773. An increase in the ratio from 0.67 to 0.77 gives an increase in the stress experience (SISQ) of 0.91 units (on a 1–5 scale); an increase from 0.67 to 0.72 gives an increase of 0.455 units, and so on.

### 3.3. The Intra-Individual Variability

Table 4 displays odds ratios (OR) for SMS series 1 and 2, for both A (i.e., using means (M) and standard deviations (SD)) and B (i.e., when taking into account the rate of change by using first difference).

The number of individuals in the subgroups in SMS 1 series was 31 (39%), 18 (22.5%), 21 (26%) and 10 (12.5%) for the LL (low M/low SD), LH (low M/high SD), HL (high M/low SD), and HH (high M/high SD) subgroup respectively. For SMS 2 series, the number of individuals was 38 (47,5%), 14 (17,5%), 15 (19%) and 13 (16%) for the LL, LH, HL and HH respectively.

Our results show no statistical difference between the four stress subgroups (LL, LH, HL and HH) for SMS 1 series regarding age, number of children in the home, years working in the organization and overtime (information not displayed). The LL subgroup (low M/low SD) was used as a comparison group. This was true in both A and B. There was a significant difference between subgroups regarding baseline measurement levels of over-commitment, exhaustion and depression, as shown in Table 4. For every increase in the score on the depression, exhaustion or over-commitment scale, the odds of being in the HL subgroup increased significantly, both when using usual M and SD and when taking into account the rate of change.

Results for the SMS 2 series showed the same pattern as for the SMS 1 series, i.e., no differences regarding the background data, but significant differences between the subgroups regarding the 6-month follow up levels of over-commitment, exhaustion and depression. For every increase in score on the depression scale, the odds of being in the HL subgroup increased significantly, both when using usual M and SD and when taking into account the rate of change. For every increase in score on the exhaustion scale, the odds of belonging to either the HL or the HH subgroup increased significantly. When taking into account the rate of change in SMS series 2 (that comprises twice as many weeks as SMS 1 series), the higher the exhaustion the higher the odds of belonging to the HH group (OR = 1.463, 95% CI: 1.180; 1.814). For over-commitment, the odds of belonging to HL or HH group were similar.

## 4. Discussion

This study provides longitudinal data about the week-by-week stress experience of a group of primary health care employees in Sweden. In addition, we explored whether their stress experience was related to objective measures of workload. We also looked at the intra-individual variability of the stress experience in a time series. To our knowledge, this is the first study to use weekly text messages for the frequent monitoring of the stress experience at work over such a prolonged period.

### 4.1. The Experience of StressOver Time

Where the weekly stress experience of this sample was concerned, there was a clear variability over time, with higher stress during the fall than in spring and summer. This underscores the importance of assessing stress at several time points, because questionnaire responses can indicate stability where such stability does not exist. We observed clear differences between the trajectories of experienced stress in the different subgroups in this sample. The stress trajectory of the high-stress subgroup has a different course and has markedly more variability than the low-stress subgroup. An explanation of these findings may be that the high-stress subgroup has a higher sensitization to the stress stimuli and increased amygdala reactivity, which is in line with other research on stress [28,49].

One explanation for the variability in stress experience in the total sample could be that people may be healthier during the spring and summer and therefore the number of patients seeking care is lower, resulting in less pressure on the health care workers. However, variability could also be due to a seasonal variation in overall workload as discussed below in relation to objective data. Moreover, in northern countries depressive symptoms peak in the winter in the general population [50] which could also affect the employees and their susceptibility to stress. As there is a three months gap between the SMS 1 and SMS 2 series some caution is in place regarding these explanations.

### 4.2. The Experience of Stress and the Objective Measures of Quantitative Workload

The objective monthly group-level workload data correlated highly with the group level stress experience. Because these workload data show that the workload is lower during the summer, it seems to confirm the above assumption about seasonal variations. The practical implications of these findings are that one possible intervention to reduce stress would be either to reduce the amount of work (by recruiting extra personnel during the busy periods caused by seasonal variation) or to empower the individuals at risk by means of individual stress management interventions, as these have been shown to be effective [51].

However, when workload is expressed as the ratio of time and objective raw data, somewhat different and important information about the relationship between workload and stress experience appears. The fewer core tasks (visits, administration, phone calls, etc.) accomplished per hour worked, the higher the experience of stress. This might suggest that other processes, possibly other group or organizational tasks not directly related to the core tasks, may be “stealing time” and act as a basis for a “hindrance appraisal” [52]. These processes might be regular meetings, development work, the introduction of new colleagues, conflicts, in-house training, and so on. In other words, these primary health care employees might experience stress because they feel inefficient (or get the statistical feedback that they are inefficient), i.e., they are not accomplishing what they planned or should have done in relation to their core tasks.

Research into job demands as a challenge or a hindrance demonstrates that what is seen as a hindrance varies according to occupational sector. Having too much work, for example, is a hindrance for nurses as it makes it more difficult to provide good care. On the other hand, for other professions such as journalists, deadlines can be a stimulus [53]. Important as it is, the existing research still relies entirely on subjective measurements and does not distinguish between the job demands posed by the core tasks and the possible workload related to other group and organizational processes. Our finding is interesting because it indicates that there is a need for further research into what may take time from the core tasks and thereby lead to a feeling of hindrance and inefficiency. Furthermore, working long hours without a feeling of accomplishment and a sense of achievement and efficacy (which our data may be indicating) could be an important part of a “perceived workload”. Perceived workload is often seen as a mediator between objective workload and ill-health [20]. The seasonal variation in the stress experience in our sample goes well with this line of thought. It is likely that during the summer fewer organizational processes such as development work, meetings, feedback on productivity, etc., will take place (at least in Sweden) and more time will be available for the core tasks, so the view of what constitutes a hindrance might change because there are fewer processes in operation.

The sparse amount of available research about how objective workload (for example work hours and patient load) affects perceived stress and burnout is ambiguous. One study found that number of work hours was associated with stress [54]. Another found that they were not a significant predictor of stress and burnout [55]. The number of work hours in the Panagopoulou et al. study [55] was assessed by the participant’s report of mean working hours per week and are thereby not objective data in a proper sense. Furthermore, the analysis used the raw data only. However, their finding that work hours are not a predictor of stress fits well with our results which show that the raw objective data may not be the only appropriate variable to correlate with stress experience or burn out. Raw objective data may also conceal the important information that becomes apparent when we use the ratios of time and raw data: perceived efficiency. As a part of Neoliberalism and New Public Management (NPM) the pursuit of efficiency is a part of a “deep transformation of culture” [56], and units and employees are accountable through performance management. Hence, the higher the ratio, the lower the accomplishment, the higher the stress. If the accomplishment on a group level is perceived and fed back as low, it might affect and drain both groups and individual employees. If lower accomplishment is due to some kind of organizational constraints, it may lead to frustration and stress [23].

### 4.3. The Intra-Individual Variability

Where intra-individual variability is concerned, our results show that people in the high- and low-stress subgroups did not differ in terms of any of the sociodemographic variables. What we notice is that people with depression or exhaustion, or people who report over-commitment, had higher odds of being in the high-stress group. This is not be surprising as prolonged stress is one of the risk factors for the development of depression and exhaustion and was not either the focus of the study. What is interesting is to look at the subgroups created using the rate of change (first derivative). The highest odds of being in the HH group (i.e., high stress and high rate of change in stress experience) were found among individuals with exhaustion. Future research will show whether high intra-individual variability in the stress experience is a risk profile for the development of exhaustion, or if intra-individual variability becomes more pronounced during the exhaustion process.

The SMS series 2 comprises more than twice as many weeks as the SMS series 1 and is therefore probably more reliable. There are some indications that people with depression have higher odds of being in the HL subgroup (high stress, low rate of change) while individuals who score higher on exhaustion have higher odds of being in the HH subgroup (high stress, high rate of change). This needs to be explored in future studies. However, linking back to the introduction section, the conditions due to prolonged stress such as depression and exhaustion may have different biophysiological markers [34] but possibly also different patterns of reacting on further stress. In this study it is exemplified by exhausted individuals reduced tolerance of further stress, i.e., reacting strongly on “everything” and illustrated by high rate of change in their stress experience.

Our explorative study was limited by the number of individuals in the four subgroups, all of which, except for the LL subgroup (low stress, low rate of change) were small. We believe therefore that bigger subgroups would give more reliable results and that this would be a valuable investigation, for example for tailoring of optimal conditions for return to work processes.

### 4.4. Methodological Considerations

The first limitation of the study is the small sample sizes of the subgroups as described above. Another limitation was that a detailed examination of the objective data, such as weekly instead of monthly workload data or the data for the different occupational subgroups, was not possible due to the difficulty of retrieving the information from the administrative system. Furthermore, the SMS 1 series consisted of only 12 weeks. We believe this contributed to the non-significant beta values in the GEE analysis, because objective data were given only monthly and therefore consisted of only three measurement points.

A fourth limitation is that the study participants had to evaluate their stress experience “these days” which gives a risk of recall bias [57] and use of heuristics in reporting [58]. We argue that, since they received SMS messages on a weekly basis, they would relate to the previous week. An alternative would have been to use the experience sampling method (ESM) [59] and ask participants also to report specific stressful events in their daily working life. We reasoned that such intensive data sampling might add yet more stress for the study participants.

Our single-item measure of stress is framed in such a way that it asks for both the general stress arousal experience and the activation of autonomic nervous system (i.e., “…a state in which a person feels tense, restless, nervous or anxious…”) and for sleep difficulties and rumination (i.e., “…unable to sleep at night because his/her mind is troubled…”). We are aware that this can be seen as a concern, but we see it as a strength as it depicts the stress experience as a complex human experience involving both physiological, emotional and cognitive components.

The repeated SMS data collection gives an opportunity to study phenomena, in this case the stress experience, close to the real world i.e., displaying ecological validity. Another advantage is that the variability of the stress phenomena can be charted, both at individual and at group level. Furthermore, it facilitates both intra-personal and inter-personal comparisons. To collect subjective experience as well as objective behavioral data (i.e., smartphone-sensing apps recording for example number of steps, heart rate, etc.) is already common in psychological research [60]. We believe that our contribution demonstrates one way of combining objective organizational data and the new technology, which in future studies could further develop work-related stress research. The results of this study also strengthen the validity of the single item stress question for the working population [39,40], because the objective organizational data can to some degree act as an uncontaminated criterion, [61] meaning they are independent of what is perceived by individuals.

## 5. Conclusions and Implication for Practice

We have shown how text messaging can be used in a work setting to track employees’ stress levels and to link them to objective workload measures. However, in future research, examining both the stress and the recovery process within the studied individuals would bring another dimension to our understanding of how stress is sustained or reduced. Proactive managers and work groups could then use this methodology to follow their own stress levels for preventive purposes and to track whether problem-solving attempts and activities are giving desirable results. At an individual level, the effects of rehabilitation and return to work activities could be tracked by means of a daily, bi-daily or weekly follow up in a simple, feasible manner. A prerequisite for the uptake of such use of new technology is close collaboration between employer, employees and the occupational health services.

## Figures and Tables

**Figure 1 ijerph-17-00680-f001:**
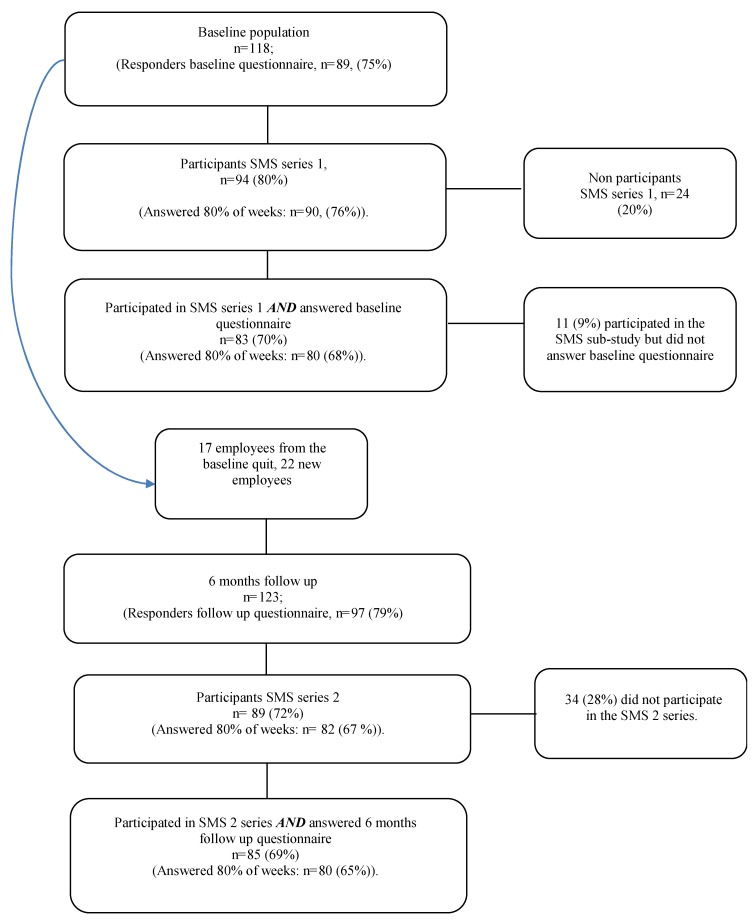
Flowchart of participants in the SMS1 and SMS2 series. To explore the experience of stress (Aim 1) and to examine group level associations between the experience of stress and objective organisational measures of quantitative workload (Aim 2), we used the data from all participants in the SMS1 and SMS2 study who answered at least 80% of weeks, regardless of if they completed questionnaires. To describe the intra-individual variability and explore subgroups we only used data from employees who BOTH participated in 80% of weeks of the SMS study AND answered questionnaires at the baseline (for SMS1) or 6-month follow up (for SMS2).

**Figure 2 ijerph-17-00680-f002:**
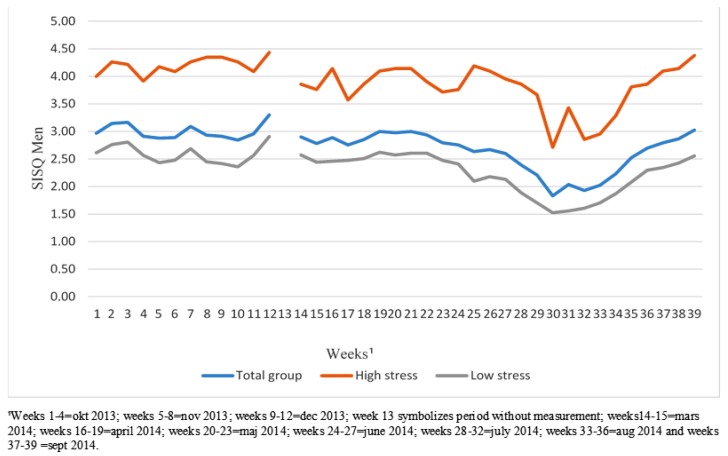
SISQ Mean for SMS series 1 and 2. Total group and high and low stress subgroups. Dichotomization at upper quartile, i.e., high stress subgroup SMS1 series = SISQ >3.67; high stress subgroup SMS2 series = SISQ >3.19.

**Table 1 ijerph-17-00680-t001:** The cut off points used in the formation of the four stress subgroups, i.e., means (M) and standard deviations (SD) based on all the stress scores for all participants in the SMS series 1 and the SMS series 2.

	SMS Series 1		SMS Series 2	
Sub-Group ^a^	M	Sd	M	Sd
LL	<3.462	<0.927	<2.999	<1.067
LH	<3.462	≥0.927	<2.999	≥1.067
HL	≥3.462	<0.927	≥2.999	<1.067
HH	≥3.462	≥0.927	≥2.999	≥1.067

^a^ LL (low M/low SD), LH (low M/high SD), HL (high M/low SD), and HH (high M/high SD).

**Table 2 ijerph-17-00680-t002:** Background data for (a) all the questionnaire responders at the baseline measurement; (b) for those who both participated 80% of weeks in the SMS 1 series and answered the baseline questionnaire; (c) all questionnaire responders at the 6 month follow up; (d) those who both participated 80% of weeks in the SMS 2 series and answered baseline questionnaire.

	Baseline(a)	SMS 1 80% of Weeks(b)	6 Month Follow Up(c)	SMS 2 80% of Weeks(d)
Variable	N = 89	N = 80	N = 97	N = 80
Sex, n (%)				
*Female*	75 (84.3)	69 (86.3)	81 (84)	69 (86)
*Male*	14 (15.7)	11 (13.7)	16 (16)	11 (14)
Age, years, mean (SD)	46.1 (11.6)	46.3 (11.6)	45.9 (11.8)	46.0 (11.7)
Working hours ^1^, mean (SD)	37.5 (5.3)	37.5 (4.9)	36.5 (6.4)	36.8 (5.3)
Overtime work ^2^, mean, (SD)	8.1 (26.8)	5.4 (7.0)	5.7 (9.4)	4.6 (7.3)
Overall health ^3^, mean (SD)	2.0 (0.8)	2.0 (0.8)	2.0 (.7)	2.0 (0.7)
Formal ed. level, n (%)				
*Comprehensive school*	-	-		-
*Secondary school*	15 (17)	15 (19)	15 (16)	12 (15)
*University education*	71 (80)	63 (79)	80 (82)	68 (85)
*Higher academic ed.*	3 (3)	2 (2)	2 (2)	-
Type of household, n (%)				
*One person household*	16 (18)	15 (19)	14 (14)	13 (16)
*Single parent*	2 (2)	2 (2)	3 (3)	3 (4)
*Couple without children*	30 (34)	27 (34)	33 (34)	27 (34)
*Couple with children*	41 (46)	36 (45)	47 (49)	37 (46.)
Years at this organization, n (%)				
*Less than 1 year*	11 (12)	8 (10)	18 (18)	11 (13.8)
*1–2 years*	22 (25)	22 (28)	15 (15)	14 (17.5)
*3–5 years*	23 (26)	20 (25)	24 (25)	21 (26.3)
*6–10 years*	14 (16)	13 (16)	18 (19)	17 (21.3)
*More than 10 years*	19 (21)	17 (21)	22 (23)	17 (21.3)
Profession, n (%)				
*Nurse*	25 (28)	23 (29)	31 (32)	25 (31.3)
*Physiotherapist*	12 (13)	12 (15)	16 (17)	15 (18.8)
*Physician*	13 (15)	10 (13)	11 (11)	7 (8.8)
*Medical secretary*	11 (12)	11 (14)	10 (10)	8 (10)
*Midwife*	8 (9)	6 (7)	7 (7)	5 (6.3)
*Laboratory technician*	5 (6)	4 (5)	6 (6)	6 (7.5)
*Assistant nurse*	6 (7)	6 (7)	6 (6)	5 (6.3)
*Counselor*	5 (6)	4 (5)	5 (5)	4 (5.0)
*Manager/Assist. Man.*	3 (3)	3 (4)	4 (4)	4 (5.0)

*Dietitian*	1 (1)	1 (1)	1 (1)	1 (1.3)

^1^ per week. ^2^ with baseline outlier. Without outlier: M = 5.0, SD = 5.8. ^3^ single question from General Health Questionnaire (GHQ) [Goldberg & Williams, 1988]. Scores 1–5.

**Table 3 ijerph-17-00680-t003:** Generalised estimating equations parameter estimates for the outcome variable monthly stress experience, as measured by a single item stress question administered weekly by means off a text message system (SMS series 2).

Variable ^1^	Total Group		
	b	95% CI	p
*Time (Hours worked)*	0.163	0.128; 0.198	0.001
*Total amount of tasks*	0.155	0.122; 0.187	0.001
*No of patient visits*	0.277	0.215; 0.339	0.001
*No of administrative tasks*	0.465	0.356; 0.575	0.001
*No of calls answered*	0.520	0.412; 0.628	0.001
*Ratio time/total tasks*	9.133	6.90; 11.36	0.001
*Ratio time/patient visits*	3.241	2.27; 4.21	0.001
*Ratio time/admin. tasks*	1.791	1.35; 2.23	0.001
*Ratio time/calls answered*	2.099	1.54; 2.57	0.001

^1^ All variables are calculated per month.

**Table 4 ijerph-17-00680-t004:** Multinomial logistic regression with subgroup of intra-individual variability as a dependent variable and over-commitment, depression and exhaustion as independent variables for SMS series 1 and 2. Subgroups created (A) using common means (M) and standard deviations (SD) and (B) using the rate of change (i.e., first derivative). (Subgroups HH = high M/high SD), HL = high M/low SD), LH = low M/high SD and LL = low M/low SD).

	SMS 1 Series (12 Weeks)			SMS 2 Series (26 Weeks)		
	OR ^a^	95% CI		OR	95% CI	
	Exp(B)	Lower	Upper	Exp(B)	Lower	Upper
*Variables*	(A) Subgroups created using common (M) and (SD)					
*Over-commitment* ^1^						
*LH*	**1.186**	1.007	1.397	1.082	0.926	1.265
*HL*	**1.345**	1.137	1.592	**1.249**	1.064	1.465
*HH*	1.038	0.843	1.277	**1.233**	1.045	1.455
*Depression* ^2^						
*LH*	1.104	0.841	1.449	0.885	0.620	1.262
*HL*	**1.386**	1.087	1.766	**1.360**	1.090	1.697
*HH*	0.972	0.662	1.428	1.142	0.895	1.456
*Exhaustion* ^3^						
*LH*	1.167	0.994	1.371	1.101	0.916	1.324
*HL*	**1.490**	1.231	1.804	**1.371**	1.125	1.670
*HH*	0.984	0.806	1.201	**1.350**	1.102	1.653
	(B) Subgroups created using the rate of change (first derivative)					
*Over-commitment* ^1^						
*LH*	0.933	0.794	1.096	0.997	0.852	1.167
*HL*	**1.189**	1.028	1.375	**1.189**	1.016	1.391
*HH*	1.004	0.840	1.199	**1.234**	1.050	1.451
*Depression* ^2^						
*LH*	1.101	0.842	1.439	0.977	0.726	1.315
*HL*	**1.351**	1.064	1.715	**1.285**	1.026	1.609
*HH*	1.103	0.809	1.503	**1.278**	1.020	1.600
*Exhaustion* ^3^						
*LH*	0.951	0.821	1.102	1.094	0.924	1.296
*HL*	**1.269**	1.090	1.477	**1.283**	1.054	1.562
*HH*	1.028	0.869	1.216	**1.463**	1.180	1.814

The reference category is LL = Low M/low SD. ^a^ Significant Odds Ratio (OR) in bold. ^1^ Measured by OLBI [41]. ^2^ Measured by HAD [44]. ^3^ Measured by the Over-commitment subscale of the ERI-questionnaire [Siegrist, 1996, 2013].

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
