# Peer review of "Experience of Stress Assessed by Text Messages and Its Association with Objective Workload—A Longitudinal Study"

_ijerph, 2020, doi:10.3390/ijerph17030680_

Round 1
Reviewer 1 Report
The authors develop a study to Exploring stress trajectories in detail.The objective was to describe their experience of stress, explore the intra-individual variability and examine the association between the experience of stress and the objective workload.
The manuscript is well organized nonetheless it is necessary there is comparison with the increase in the global research output in the field and include all relevant references: ( Aiken et al,).
The reader should be convinced why this study is important. I believe the results are interesting to hospital managements and health care policy takers.
design
The objectives of the study are clearly defined.
In my opinion The methods are clear and the ethical standards are guaranteed.
Nonetheless it is not made clear the focus group and the systems the sms.
Although the results are limited to a small samples, the importance of the factors analyzed and the significance of the correlations found justify publication.
The first sentence of conclusion is not a conclusion! They need improved the conclusion.
Author Response
Answer to Reviewer 1:
We would first like to thank you for your time and effort.
Comments and Suggestions for Authors
The authors develop a study to Exploring stress trajectories in detail.The objective was to describe their experience of stress, explore the intra-individual variability and examine the association between the experience of stress and the objective workload.
The manuscript is well organized nonetheless it is necessary there is comparison with the increase in the global research output in the field and include all relevant references: ( Aiken et al,).
The reader should be convinced why this study is important. I believe the results are interesting to hospital managements and health care policy takers.
Thank you. We have now added the following text to the introduction (page 2, lines 60-71):
“Furthermore, different aspects of the work environment have an impact on employee wellbeing, but also on the quality of patient care. For example, a previous study has shown that high workload was associated with less individualized patient care [23]. In other words, less consideration was given to the patients´ needs and preferences due to high workload. Moreover, Aiken et al. [24] showed that hospitals with better care environments had lower risk of patient death and failure to rescue than hospitals with poor care environments. A literature review [25] showed that nurse staffing (most often measured by calculating nurse to patient ratio) was associated with better patient outcomes both in general care populations and in intensive care units. More detailed inquiry into the association between objectively measured workload and employee stress and health should therefore be of interest for the hospital managements, primary health care managements and health care policy decision makers.
design
The objectives of the study are clearly defined.
In my opinion The methods are clear and the ethical standards are guaranteed.
Nonetheless it is not made clear the focus group and the systems the sms.
Although the results are limited to a small samples, the importance of the factors analyzed and the significance of the correlations found justify publication.
Thank you for your feedback. We have now changed our introduction in several paragraphs. We have also clarified which population was studied as well as added information about the SMS system (page 4, lines 160-171).
The first sentence of conclusion is not a conclusion! They need improved the conclusion.
We agree. We have now moved the first three sentences from the conclusion to the introduction section (page 2, line 45-48).
Reviewer 2 Report
The main purpose of this work is to obtain data through SMS and analyze the relationship between stress and workload. To achieve this, the author used a different survey method from the traditional one, Weekly text messages (short message service, SMS). Overall, the structure of this paper is clear.
However, I think the following Suggestions are meaningful for further improvement:
In line 136, the author mentions the SPA study for the first time and expresses it with a capital letter abbreviation, without explaining what the abbreviation means. I think it is possible to add an explanation so that readers can clearly understand what the SPA research refers to.
2. In line 372, "an increase from 0.67 to .072 gives an increase of 0.455 units, and so on", is there a writing error?
Author Response
Answer to Reviewer 2:
We would first like to thank you for your time and effort.
Comments and Suggestions for Authors
The main purpose of this work is to obtain data through SMS and analyze the relationship between stress and workload. To achieve this, the author used a different survey method from the traditional one, Weekly text messages (short message service, SMS). Overall, the structure of this paper is clear.
However, I think the following Suggestions are meaningful for further improvement:
In line 136, the author mentions the SPA study for the first time and expresses it with a capital letter abbreviation, without explaining what the abbreviation means. I think it is possible to add an explanation so that readers can clearly understand what the SPA research refers to.
Thank you for your feedback and the suggestions you made.We have now clarified in the text what the abbreviation SPA stands for: Stress prevention at work (page 4, line 160-161 in the revised manuscript)
In line 372, "an increase from 0.67 to .072 gives an increase of 0.455 units, and so on", is there a writing error?Thank you for this comment, it should indeed be .72 and NOT .072. This has now been changed in the manuscript (see page 12, line 358 in revised manuscript)
Reviewer 3 Report
The work is fine. However literature is a little biased towards only Nordic countries studies. Authors should quote and discuss also other studies including doi: 10.2147/PRBM.S173282, doi: 10.2196/13154, 10.2147/PRBM.S169918, 10.2196/mhealth.9186, 10.2147/PRBM.S41386.
Author Response
Answer to Reviewer 3
We would first like to thank you for your review.
Comments and Suggestions for Authors
The work is fine. However literature is a little biased towards only Nordic countries studies. Authors should quote and discuss also other studies including doi: 10.2147/PRBM.S173282, doi: 10.2196/13154, 10.2147/PRBM.S169918, 10.2196/mhealth.9186, 10.2147/PRBM.S41386.
Thank you for your revision and suggestion. We have now added more references in the introduction section (see page 2, lines 45-48, 59-71 and page 3, 115-125 in the revised manuscript).
However, we were uncertain if the doi number you suggested was correct? (As the paper we got access to was about nomophobia, which is outside the focus of this study?)
Reviewer 4 Report
The authors conduct a longitudinal study on individuals' stress investigating on the time series trajectory data, association with workload, and intra-individual variability. I appreciate than the authors conducted the study with text messaging that maintained high response rate and provided the opportunity to anlayze the data with diverse perspectives. I believe that the subject has contribution in the field, however, needs to be revised for the following reasons.
The contribution of the paper is not clear in the introduction. The authors emphasize the use of SMS over questionnaires for stress level assesement. I believe that it is beneficial for instant and ease of use for responses which is valuable for these types of studies. However, as the paper is not about comparing SMS vs. questionnaires based studies, it should be toned down. The emphasis on gathering weekly data can be mentioned as a benefit. I think the intro should focus more on that part, especially how previous longitudinal studies were conducted and comparise to show the novelty on how frequent data was collected or what was analyzed.
The analysis is too narrative and needs more sound statistical analysis.
- Section 2.3.1: The rational about how they divide the high- and low- stress sub-groups is not clear. The authors seperated SMS1 and SMS 2 group for the calculation for the upper quartile. It means that consider the group differently, if they consider the group to be homogeneous than the threshold should be consistent by calcuating the data as a whole. It they could try both and mention that there is no difference.
- Section 2.3.2: The authors conducted a monthly calculation. The stree level survey was conducted weekly. It should be clarified why montly. Especially for SMS1, 12 weeks would be summarized to 3 months. And in the result section, only SMS 2 group is reported.
- Section 2.3.3: I am not sure about the grouping methodology for the 4 groups. I would approach is first for 2 groups simply. There should be more validation on the grouping method.
The points mentioned above affect the validisty for Section 3.
More details and additional anlaysis is required for the result section.
- Section 3.1: Fig 2. SD should be plotted on the graph. There is cleary a seasonal effect, however, the month for SMS1 and SMS2 do not overlap. There could additional effects which should be explicltly mentioned.
I also suggest to have subsections for the Discussion section. This will help the audience to grasp the points.
Others
-3.1 section number should be 4.1
- More literature on the the frequency monitoring gathering information
Author Response
Answer to Reviewer 4
We would like to thank the reviewer for the time spent on this review and the comments given.
The authors conduct a longitudinal study on individuals' stress investigating on the time series trajectory data, association with workload, and intra-individual variability. I appreciate than the authors conducted the study with text messaging that maintained high response rate and provided the opportunity to anlayze the data with diverse perspectives. I believe that the subject has contribution in the field, however, needs to be revised for the following reasons.
The contribution of the paper is not clear in the introduction. The authors emphasize the use of SMS over questionnaires for stress level assesement. I believe that it is beneficial for instant and ease of use for responses which is valuable for these types of studies. However, as the paper is not about comparing SMS vs. questionnaires based studies, it should be toned down. The emphasis on gathering weekly data can be mentioned as a benefit. I think the intro should focus more on that part, especially how previous longitudinal studies were conducted and comparise to show the novelty on how frequent data was collected or what was analyzed.
We have in accordance with the reviewer´s comments “toned down” the introduction and made the following changes (page 1, lines 34-39):
Typically, stress level assessment is carried out by questionnaires distributed at baseline and at follow up [3]. Questionnaires consisting of several items are preferable from a psychometric point of view, as factor analysis can be used to measure their internal structure [4]. However, they are not always suitable for frequent and prolonged monitoring and may sometimes be burdensome for participants. Moreover, collecting data by weekly messages provides more information about the trajectories of stress (given that the single items used are validated).
We also followed your recommendations about emphases on weekly data and have elaborated on two relevant earlier studies (See page 2 lines 59-71, but also 45-47.)
The analysis is too narrative and needs more sound statistical analysis.
- Section 2.3.1: The rational about how they divide the high- and low- stress sub-groups is not clear. The authors seperated SMS1 and SMS 2 group for the calculation for the upper quartile. It means that consider the group differently, if they consider the group to be homogeneous than the threshold should be consistent by calcuating the data as a whole. It they could try both and mention that there is no difference.
We understand that parts of the text were not described clearly. We have therefore clarified the text by making the following changes in this paragraph page 7, (lines 242-254):
“The sample here consisted of all employees who responded to the SMS at least 80% of the weeks, regardless of whether they completed the questionnaires. The mean of the SISQ was computed for all the participants in SMS series 1 (12 weeks) and SMS series 2 (26 weeks). Based on the present sample data (frequency tables), the cut off for the dichotomization of high and low stress sub-groups was set at the upper quartile (SISQ mean ≥ 3.67 = high stress subgroup in SMS series 1 and SISQ mean ≥ 3.19 = high stress subgroup in SMS 2). The reason for this was that we were interested in the trajectories of the most stressed employees. Because we had a 1-5 scale, the cut off had to be higher than three. The quartile cut-off gave the acceptable number of individuals (25 of 90 employees in the SMS 1; 21 of 82 employees in SMS series 2). We then computed means and standard deviations for every week for subgroups with high and low stress for SMS series 1 and 2 separately. As there was a small variation in the number of employees participating in the two SMS series (due to the natural turnover described above) and a three months gap between the two series, we could not treat the two SMS series as one.”
- Section 2.3.2: The authors conducted a monthly calculation. The stree level survey was conducted weekly. It should be clarified why montly. Especially for SMS1, 12 weeks would be summarized to 3 months. And in the result section, only SMS 2 group is reported.
Thank you. We agree and have now clarified why monthly calculation were made. We understand that the wording in the paragraph was confusing and have therefore added the following paragraph (lines 257-259):
“The objective workload data were obtained monthly per health care unit and aggregated for the total sample. It was not possible to access the objective data on a weekly level and therefore the monthly SISQ mean for every individual was calculated from their weekly stress ratings.”
Regarding the reviewers comment of not reporting SMS 1, this decision was made because no (significant) association was observed between the monthly experience of stress and quantitative workload.
Therefore, we did not feel it was necessary to display this data.
Here are parameter estimates for ratios as an example:
|
Variable¹ |
b |
95% CI |
p |
|
Ratio time/total tasks |
-.2.239 |
-5.824; 1.345 |
.221 |
|
Ratio time/patient visits |
.021 |
-1.712; 1.755 |
.981 |
|
Ratio time/admin. tasks |
-.214 |
-.900; .472 |
.541 |
|
Ratio time/calls answered |
-.656 |
-1.809; .495 |
.264 |
¹All variables are calculated per month.
- Section 2.3.3: I am not sure about the grouping methodology for the 4 groups. I would approach is first for 2 groups simply. There should be more validation on the grouping method.
The points mentioned above affect the validisty for Section 3.
Thank you for your comment. Our reasoning concerning this is as follows:
To our knowledge, this is the first study looking at the rate of change in repeated weekly stress experience. In other words, it is an exploratory study. We know that Mean and Std are important, however we do not know anything about rate of change in stress experience in different populations and therefore different combinations of mean and rate of change could be possible. For example, even an individual with low mean stress could oscillate fast from week to week between say 1 and 3 on a 1-5 scale, while others are more “stable”, “low mean” individuals. Here is an example of two individuals with similar Mean and Std but different rate of change, which we hope will help explain our point:
PLEASE SEE ALSO THE ATTACHMENT, for some reasons we could not paste the figure here...
We hope more research will follow looking at this kind of grouping method for different populations.
More details and additional anlaysis is required for the result section.
- Section 3.1: Fig 2. SD should be plotted on the graph.
Thank you for your comment. When plotting the SD on the graph, the figure unfortunately became very unclear. A decision was therefore made to have Appendix A with Descriptive data, that display both SISQ Means and Standard deviations for both SMS 1 and SMS 2 series.
There is cleary a seasonal effect, however, the month for SMS1 and SMS2 do not overlap. There could additional effects which should be explicltly mentioned.
We followed your advice and have added information regarding this to the manuscript. See page 11, lines 342-343 in the results section, as well as discussion section, page 15, lines 417-418.
I also suggest to have subsections for the Discussion section. This will help the audience to grasp the points.
We have added subsections as recommended.
Others
-3.1 section number should be 4.1
Thank you. It is now 4.4
- More literature on the the frequency monitoring gathering information
We are not completely sure about this comment. However, some literature of collecting frequent data is now described in the revised paper, on page 2, line 59-71.

Round 2
Reviewer 4 Report
The revision properly incorporates the comments. The article is ready to be published.